# Patterns in Temporal Networks with Higher-Order Egocentric Structures

**DOI:** 10.3390/e26030256

**Published:** 2024-03-13

**Authors:** Beatriz Arregui-García, Antonio Longa, Quintino Francesco Lotito, Sandro Meloni, Giulia Cencetti

**Affiliations:** 1Instituto de Física Interdisciplinar y Sistemas Complejos IFISC (CSIC-UIB), Campus UIB, 07122 Palma de Mallorca, Spain; 2DISI Department of Information Engineering and Computer Science, University of Trento, 38123 Trento, Italy; antonio.longa@unitn.it (A.L.);; 3Aix-Marseille Univ, Université de Toulon, CNRS, CPT, 13009 Marseille, France

**Keywords:** temporal hypergraphs, social interactions, motifs

## Abstract

The analysis of complex and time-evolving interactions, such as those within social dynamics, represents a current challenge in the science of complex systems. Temporal networks stand as a suitable tool for schematizing such systems, encoding all the interactions appearing between pairs of individuals in discrete time. Over the years, network science has developed many measures to analyze and compare temporal networks. Some of them imply a decomposition of the network into small pieces of interactions; i.e., only involving a few nodes for a short time range. Along this line, a possible way to decompose a network is to assume an egocentric perspective; i.e., to consider for each node the time evolution of its neighborhood. This was proposed by Longa et al. by defining the “egocentric temporal neighborhood”, which has proven to be a useful tool for characterizing temporal networks relative to social interactions. However, this definition neglects group interactions (quite common in social domains), as they are always decomposed into pairwise connections. A more general framework that also allows considering larger interactions is represented by higher-order networks. Here, we generalize the description of social interactions to hypergraphs. Consequently, we generalize their decomposition into “hyper egocentric temporal neighborhoods”. This enables the analysis of social interactions, facilitating comparisons between different datasets or nodes within a dataset, while considering the intrinsic complexity presented by higher-order interactions. Even if we limit the order of interactions to the second order (triplets of nodes), our results reveal the importance of a higher-order representation.In fact, our analyses show that second-order structures are responsible for the majority of the variability at all scales: between datasets, amongst nodes, and over time.

## 1. Introduction

Networks, characterized by interconnected nodes and edges, have become essential tools for modeling a multitude of phenomena, ranging from social interactions to biological and physical systems [1,2,3].

While they have proven to be a valuable framework, networks frequently fail to capture the entire complexity of real-world systems. For example, they can only capture static interactions, while many real-world interactions tend to be dynamic; i.e., they are established and destroyed during the life of a system. These nuances are better captured by more sophisticated frameworks, such as temporal networks [4,5], in which edges are labeled with timestamps representing their temporal activations. We use temporal networks to describe the synchronous activation of remote areas of the brain [6], social networks [7], blockchain [8], mobility [9], communications [10], and the rapidly varying physical interactions between individuals in a closed environment [11,12,13].

The flexibility of temporal networks in describing time-varying interactions comes at the cost of a more complex representation and analysis. Several tools have been proposed in the literature to cope with the complexity of adding the temporal dimension to network analysis, including methods for community detection [14,15], dense subgraph discovery [16,17,18,19], surrogate network generation [20], and visualization [21]. Despite such complexity, capturing more fine-grained information about the temporal properties of networks is fundamental, as temporal links affect the dynamics in networks, including random walks [22], epidemics [23,24], and other diffusion [25] and evolutionary [26] processes.

To identify relevant structures in temporal networks and describe node behavior through time, it could be useful to decompose networks into small pieces [27,28,29,30,31,32,33], i.e., a few nodes and their connections for limited temporal snapshots [28,29], and to use these repeated patterns to characterize and compare different networks [34]. Along this line, one possible strategy is to assume a so-called *egocentric perspective*, as recently proposed by Longa et al. [35]. This approach consists of considering the evolution of the neighborhood of a node for a small number of temporal layers. These small temporal subgraphs can be collected for every node of a network and at every time step. The entire set of egocentric temporal neighborhoods (ETN) can be reduced to a smaller set of only significant structures with respect to a null model, egocentric temporal motifs (ETM). ETNs and ETMs allow for the efficient identification of repeated interaction patterns among individuals in social settings. The ego perspective indeed simplifies motif identification by comparing egocentric temporal sub-networks using a signature, a bit vector, thus avoiding the computational complexities associated with standard motif mining (which usually includes the graph isomorphism problem). However, a drawback of this approach is that it neglects connections among the neighbors of the ego node at each timestamp, thus destroying correlations in the creation of links between different ego nodes, a feature of paramount importance, for example, in temporal social interactions [36].

In recent years, and to take into account such more complex structures, a new branch of network science has emerged with the study of higher-order networks [37,38]. In this approach, mathematical tools like hypergraphs can be used to represent interactions that can involve any number of nodes at the same time, effectively extending the concept of edges to hyperedges. The number of involved nodes defines the order of the hyperedge: first order for pairs of nodes, second order for triads, and so on. Recently, considerable attention has been dedicated to the exploration and understanding of hypergraphs, from microscale [31,32,33] and mesoscale properties [39,40,41,42,43,44], to centrality [45,46], hyperedge overlap [47], and clustering measures [48,49]. Higher-order interactions have been proven to affect a variety of dynamical processes in networks; for instance, random walks [50], spreading [51], and synchronization [52]. Moreover, temporal features of group interactions have been investigated under different lenses [36,53,54,55,56,57,58]. Finally, hypergraphs have found applications in describing, for instance, scientific collaborations [59], relations among species in ecosystems [60,61,62], cognitive associations [63], and simultaneous group interactions in social environments [36,64], among others.

Given these two different views, the node egocentric on one side and the higher-order on the other, the idea of combining them naturally arises. We therefore investigate how to characterize temporal networks via higher-order egocentric structures. In order to do this, we generalize the concept of ETN from graphs to hypergraphs, defining the hyper egocentric temporal neighborhood (HETN). This new framework will allow taking into account more complex information, including hyperedges and their time evolution. We analyzed 10 datasets of social interactions. To give a thorough representation of such complex and time-evolving systems, we made use of temporal hypergraphs with hyperedges up to the second order (larger interactions were decomposed into groups of three nodes). HETNs encode a description of the temporal hypergraph at the micro-scale with an egocentric perspective. We used them to compare different datasets and different nodes inside a dataset. Our results make manifest the importance of the second order of interactions in describing temporal dynamics, as they are responsible for a large part of the variability observed at all scales: between datasets, amongst individuals, and even for the same individual, over time.

## 2. Methods

### 2.1. Hyper Egocentric Temporal Neighborhood

We define a *temporal hypergraph* H=(V,E) as an ensemble comprising a set of nodes *V* and a set of temporal hyperedges *E* that encode the higher-order interactions between nodes and the time at which they take place. Time is discrete and represented by natural numbers. For simplicity, we consider hyperedges up to the second order of interactions, i.e., involving three nodes at maximum. The hyperedges of the first order can be represented as triplets (i,j,t), where i,j∈V and t∈[0,T]. Hyperedges of the second order are depicted as quadruplets (i,j,k,t), where i,j,k∈V. Here, *t* denotes the time at which the interaction takes place and *T* corresponds to the time of the last temporal layer in the hypergraph.

A *temporal hypergraph snapshot* is a static graph that represents one temporal layer of H and includes only interactions taking place at a specific time. It is defined as $H_{t}=\left(V,E_{t}\right)$, where $E_{t}$ is the set of hyperedges (i,j) and (i,j,k) that are associated to time *t*.

We can now define the *hyper egocentric temporal neighborhood* (HETN), by extending to hypergraphs the concept introduced by [35] for simple graphs.

To do that, we first need to define the *hyper egocentric neighborhood* of a designated node v∈V; i.e., the ego node, as the subgraph $H_{t}\left(v\right)$ of $H_{t}$ comprising all hyperedges that include *v* at time *t*; i.e., first-order edges (v,j) and second-order edges (v,j,k).

Thus, a HETN of time-length *k*, $H_{t}^{k}\left(v\right)$, is defined as the sequence of *k* temporal hypergraph snapshots starting from time *t*: $\left\{H_{t}\left(v\right),H_{t+1}\left(v\right),\ldots ,H_{t+k}\left(v\right)\right\}$. In a HETN, every node is linked to its next occurrence (if any) in the sequence (see Figure 1a). In other words, $H_{t}^{k}\left(v\right)$ is a temporal subgraph of H including the higher-order neighborhood of node *v* for *k* temporal layers starting from time *t*. In the rest of this work, following the analyses performed in [35], we set the time-length of the HETN to k=2, i.e., $H_{t}^{2}\left(v\right)$, to reduce the complexity of the signatures and thus produce a concise representation of the datasets.

### 2.2. Hyper Egocentric Temporal Neighborhood Signatures

All the HETNs of a temporal hypergraph can be encoded into binary signatures. Extending the algorithm proposed in [35], the encoding at first order consists in assigning, at every time (t,t+1,…,t+k), a 1 if the link between the node *v* and a neighbor *j* exists at that time or a 0 otherwise. This results in a binary key for every node $j\neq v\in H_{t}^{k}\left(v\right)$. The extension of the encoding to the second order is performed by keeping track of the triangles (second-order interactions) in which the ego node *v* participates. A binary key is assigned for every possible dyadic interaction between nodes other than the ego node, giving a total of n(n−1)/2, where *n* is the total number of nodes in the HETN excluding the ego node. In Figure 1, for example, we can see how the key encoding the temporal interaction between the ego node and node *C* is “011”, as the edge (v,C) does not exist at time *t* but exists at t+1 and t+2. A similar case is found for the triangle composed of the ego node and vertices *B* and *C*, which would be encoded by the key associated with the edge (B,C) and which results in “011”.

Once we have the binary keys encoding the dyadic (first order) and triangle interactions (second order), we proceed to defining the key for the particular HETN by sorting lexicographically the first order encoding. The new sorting at the first order will define the sorting of the second order. For the particular case shown in Figure 1, we would have 011−111−111−111−111(C−A−B−D−E) and consequently 100−000−000−111−011−000−000−000−000−001 (CA−CB−CD−CE−AB−AD−AE−BD−BE−DE). Finally, we will concatenate the first- and second-order keys: 011−111−111−111−111−−100−000−000−111−011−000−000−000−000−001. This vector is named the *hyper egocentric temporal neighborhood signature* (HETNS).

This encoding method shows its limitations when encountering isomorphic subgraphs. In fact, while the main advantage of encoding at first order is that it bypasses the problem of isomorphic HETNs [35], this is not always possible at the second order. In fact, when there is a multiplicity of keys at the first order, several encodings may be present at second order. For instance, the example in Figure 1 could also have been sorted as C−A−B−E−D (and 22 other extra combinations) at first order, resulting in a different second-order key for the very same HETN. Thus, in the event that there is a multiplicity of keys at the first order, every possible sorting combination must be checked at the second order, in order to index isomorphic HETNs under the same, unique, HETNS. In theory, this task becomes rapidly computationally unfeasible, since every element duplicated *n* times means n! possible sort combinations. However, in practice, given the sparseness of interactions of the datasets and aggregation times (see the datasets and hypergraphs subsection, below) considered, the number of non-computable cases is negligible. This allows us to compute the HETNS of each dataset in a few minutes.

A binary signature allows storing a high number of HETNs. This, in turn, lets us characterize any node in a temporal hypergraph or the entire temporal hypergraph as a whole. Indeed, to characterize a single node *v* according to this algorithm, it is sufficient to build the vector of frequencies of every HETN where *v* participates as the ego node. These can be obtained for each time *t* using a sliding window over time. Instead, to describe a whole temporal hypergraph, we analyze the HETN of all nodes with their frequencies together. We denote these HETN-based descriptions as node (hypergraph) embedding, *EMB*(*v*)* *(*EMB *(*H*))**.

Using this notation, we can consider all the HETNSs as the basis of a vector space and their frequencies as the coordinates. This allows the comparison of two nodes (hypergraphs) $v_{1}$ and $v_{2}$ ($H_{1}$ and $H_{2}$) by computing a cosine distance between the embedding vectors as
(1)$$dist\left(X_{1},X_{2}\right)=\left(1-\frac{\mathrm{EMB}\left(X_{1}\right)\cdot \mathrm{EMB}\left(X_{2}\right)}{\|\mathrm{EMB}\left(X_{1}\right)\|\;\|\mathrm{EMB}\left(X_{2}\right)\|}\right)$$
where *X* represents either a node *v* or a hypergraph H.

### 2.3. Hyper Egocentric Temporal Motifs

The set of HETNs of a hypergraph can include very basic structures with high frequency, which are common to all hypergraphs, as we will see below. For some applications, it is hence useful to filter them out and reduce the analysis to a smaller set of HETNs that are significant for describing that specific hypergraph. This filtering procedure makes use of a null model that assumes no temporal correlations between the interactions. We obtain a null model $\bar{H}$ of H by randomly shuffling the hypergraph temporal snapshots $\left\{H_{1},H_{2},\ldots ,H_{T}\right\}$ in which H may be split.

We consider HETNSs from H as significant when they satisfy the following requirements:Over-representation: $P({\bar{N}}_{\bar{H}}>N_{H})<\alpha$;Minimum deviation: $N_{H}-{\bar{N}}_{\bar{H}}\geq \beta {\bar{N}}_{\bar{H}}$;Minimum frequency: $N_{H}\geq \gamma$.

where $N_{H}$ is the number of occurrences of a HETNS in H, ${\bar{N}}_{\bar{H}}$ is the average number of occurrences of the same HETNS in $\bar{H}$, and where we set α=0.01, β=0.1 and γ=5, respectively. We refer to the resulting significant HETNS as a *hyper egocentric temporal motif* (HETM). As in the HETNS case, we denote as $EMB_{HETM}\left(H\right)$ the embedding vector describing the graph H based on their HETM frequencies. Frequencies are normalized by the maximum frequency in each vector.

### 2.4. Datasets and Hypergraphs

To validate our algorithm, we evaluated it across ten proximity contact networks. Nine of them modeled face-to-face interactions and were collected using wearable proximity sensors (radio-frequency identification (RFID) tags) during the SocioPatterns project http://www.sociopatterns.org/ (accessed on 1 December 2023). The devices recorded face-to-face interactions between users, and the data were made available as pairwise interactions with a temporal resolution of 20 seconds. Table 1 shows the number of nodes, the number of edges, and the durations (in days) for each of the following datasets:Three high schools, containing interactions between students and professors, gathered at a high school in Marseille (France) in the years 2011, 2012, and 2013 [65,66]. Metadata inform about the class of each student or the role of the professors.A primary school, containing interactions between students and teachers, collected in a French primary school [67,68].Two workplaces, containing interactions between co-workers in an office building in France in 2013 and 2015 [69,70].A hospital, containing interactions between workers and patients, collected in the geriatric ward of a university hospital in Lyon (France) [71]. Metadata inform about the role of each individual (medical doctors, patients, nurses, and administrative staff).A scientific conference, containing interactions between the attendees [70].A primate dataset, containing the interactions between Guinea baboons living in an enclosure at the CNRS Primate Center in Rousset-sur-Arc (France) [72].

In addition, we also analyzed a dataset of proximity contacts collected via Bluetooth technology. Here, the interactions depended only on physical distance and did not always imply an active interplay among people. These data were gathered at a university campus (at the Technical University of Denmark, DTU), where a group of freshman students participated for a month using phones that provided physical proximity information using Bluetooth tech [73]. We analyzed a sample of these data comprising just one week of interactions.

All these datasets come as a list of temporal pairwise interactions. Each term is encoded as (i,j,t), where *i* and *j* represent individuals, i.e., the nodes of our hypergraphs, and *t* is the time at which the interaction took place. To build a hypergraph from these data, we need to choose a temporal interval Δt that would become the time resolution of the temporal hypergraph. The timestamps appearing in the dataset were hence divided into slots of length Δt and all the interactions that took place between *t* and t+Δt were stored as interactions taking place at time *t* in the hypergraphs. We called Δt
*aggregation time*. Moreover, we divided interactions into first- and second-order; i.e., if in an interval there were the interactions (i,j), (j,k), and (k,i), we considered this a unique second-order hyperlink (i,j,k). We encoded this interaction as a hyperedge of second order, instead of three first-order edges. Larger values of Δt resulted in a larger number of interactions being included in each temporal snapshot, thus also increasing the probability of forming second.order interactions. While we considered the idea of extending the analysis to third or higher order interactions, we found that across the social contact data examined, the number of simultaneous interactions involving four or more agents diminished rapidly. Therefore, in order to make a comparison between datasets with statistically significant samples, we limited the analysis to second-order effects only.

In the following, we show the results obtained for several temporal hypergraphs generated from the above datasets using different aggregation times Δt. We start by comparing the different datasets using the distance metric defined in Equation (Equation 1) based on their $EMB_{ETM}\left(H\right)$. Afterwards, we zoom in on individual nodes and compute their HETNS. We then compute the distances among them, obtaining distance matrices that we represent using a multidimensional scaling technique. This is a dimensionality-reduction technique that consists of translating the values of a distance matrix into an *n*-dimension representation by preserving the distances in the *n*-dimension space. In our case, we use an Euclidean representation. This technique allows visualizing similarities among nodes computed from their HETNS composition, while also pointing out the different roles of the nodes given by the contact metadata. We analyze the node’s HETNS nature with a focus on distinguishing the contribution provided by the pure first- and pure second-order HETNS. Finally, we look at the temporal coherence of a node with itself by also looking at its HETNS embedding at different times.

## 3. Results

We start our analysis with HETMs, which only include the significant structures with respect to the null model. As expected, the number of distinct HETMs in real temporal hypergraphs varies with datasets and aggregation time, as shown in Figure 2a. We can observe that from all the distinct HETMs appearing for each dataset (continuous lines), the number of second-order HETMs (dashed lines) is quite large, even for low aggregation times, and this calls for a deeper analysis. We can observe that increasing the aggregation time makes the number of different HETMs increase for both first and second orders. Considering larger aggregation times indeed implies that a larger number of nodes interact with the ego node, thus increasing the complexity and hence the variability in the HETM structures. The abundance of each HETM, i.e., the number of times that a HETM appeared in the network, is reported in Figure 2b for the *Hospital* dataset. Again, although, for each aggregation time analyzed, the first four most frequent HETMs only involved pairwise interactions, starting from the fifth first and second-order HETMs seemed to be equally likely. Finally, in the following analyses, we choose a small aggregation time (2.5 min), because we recall that choosing a value of Δt implies considering interactions taking place in that range as simultaneous, which can only be considered a good approximation for ranges of a few minutes.

The set of HETMs obtained from a temporal hypergraph schematizes the behavior of the nodes of that specific hypergraph with respect to their neighborhood. Then, they can be used to characterize and compare different datasets. To do so, we defined the distance between two datasets as the inverse of the cosine similarity between the two corresponding vectors of HETM frequency embeddings (see definition in Methods). Figure 3 shows the matrix of pairwise distances between all the considered datasets. The top panel reports the distances computed considering all the HETMs, while the bottom panels were obtained by only using purely first- and purely second-order HETMs, respectively.From the top panel, we can notice a slight clusterization between datasets of the same kind; for instance, the three high school datasets showed a higher similarity among them than with the other datasets, and the same happened to the two workplaces. We can indeed suppose that people inside similar environments (like students at schools) tend to follow similar patterns of behavior. The Bluetooth dataset instead appeared more distant from the others, probably because of the different technology that implied a larger variety of interactions. Bluetooth data, in fact, record the co-location between individuals, not only face-to-face, so a recorded connection between two nodes means that the corresponding individuals are in the same area, but not necessarily interacting. This resulted in larger groups and hence the appearance of more complex HETNs than for the other datasets. Interestingly, distances at the first order were smaller with respect to considering both types of interactions, suggesting that first-order HETMs were very similar between datasets. On the contrary, the second-order HETMs showed generally larger distances among datasets and accentuated the distance of the Bluetooth dataset from the others.

If the hyper egocentric temporal neighborhoods observed in one network characterize the corresponding dataset, those found for one node characterize the behavior of the corresponding individual. We hence used them to compare individuals and identify classes of behaviors. By using a distance metric analogous to the one used above, we implemented a multidimensional scaling; i.e., we positioned nodes in a 2D-space according to the reciprocal distance (see Methods). Notice that for the node analysis in (Figure 4 and Figure 5), we made use of the entire set of HETNs, not only the HETMs. In fact, the concept of HETM has been defined to characterize a hypergraph and not singular nodes; the structures that are significant for a hypergraph do not necessarily coincide with the structures that are significant for a node. Since the concept of significance for a node (or a group of nodes) is not a trivial matter, we chose to use all the collected HETNs to characterize a node.

In Figure 4, we report for two datasets (*High School’11* and *Hospital*) the multidimensional scaling of the node distances. The size of the nodes provides information on the number of different HETNs: larger nodes have a larger variety of HETNs. In addition, each node may belong to a different class (information provided by metadata), which is represented by its color. For both datasets, three panels are shown: the distances obtained comparing all HETNS, those obtained from the first order only, and those from the second order only, respectively.

From the comparison between all HETNS (first- and second-order together) for the *High School’11* dataset (lower panels of Figure 4), we can notice a central cloud containing 98% of the nodes. In this cloud, we see that nodes representing students in different classes (PC*, PSI* and PC) are well mixed, while ones corresponding to teachers are located on the edge of the cloud. Moreover, this last group has, on average, a smaller size, meaning that they presented less variety in behavior. For the *Hospital* (upper panels of Figure 4), we can observe a higher dispersion, but we can easily appreciate that the separation between different classes of nodes partially reflects the metadata. Indeed, we notice a small cluster corresponding to the patients, which is also the class showing the lowest number of different HETNs. This cluster is surrounded by that of the administrative staff and another, more dispersed, cluster composed of medics. Nurses are instead spread around, suggesting a more varied behavior, which is not characterized by a small group of specific HETNs as for the other classes.

Again, when we restrict the analysis to the first order, we can observe in both cases a behavior similar to the previous one, with a similar clustering of the nodes. The main difference is represented by the fact that the clouds are smaller, implying a generally higher similarity between individuals. On the other hand, the second-order panels exhibit more dispersed clouds, with distances approximately twice those computed for the first order. This means that the second order reveals a higher variability in temporal neighborhoods among different nodes, something that could not be appreciated if limiting the analysis to the first order.

After analyzing individual HETNSs within each dataset, it might also be instructive to investigate whether classes of behaviors appear consistently across datasets. To do that, in Figure 5, we report the multidimensional scaling for all the nodes of all the analyzed datasets, where different node colors identify different datasets. Again, we show the results for all HETNs and those for the first order only and the second order only.

By analyzing the first and second orders together, we notice that most of the nodes are concentrated in the central part of the plane; i.e., many HETNSs are shared among datasets. There, we can find the smaller and simpler structures, mainly involving the first order (as can be seen by the five most frequent ones collected in region *A* and reported at the bottom). These are common to all datasets. Moving away from that central region, more complex structures start to appear, involving the second order too. Here, the datasets started to differentiate: the two workplaces rapidly disappear (implying that only very basic structures are present in these datasets); some other datasets (*Primates*, *Conference*, *Hospital*, and the high schools) showed only a few individuals that behaved differently from the majority; while the *Primary school* and the *DTU Bluetooth* were very spread out in space, with a large variety of behaviors. In the peripheral region *B*, we notice that second-order structures, quite uncommon in region *A*, are not only present but significant, also appearing among the five most frequent HETNs.

The central panel of Figure 5, depicting the results limited to the first order, shows a different pattern, where most of the datasets are collapsed in the middle and only a few nodes are differentiated. Here, the two regions *A* and *B*, central and peripheral, show very similar and simple structures(four HETNs out of five are common to the two areas). Moreover, in both cases, the five reported structures cover more than 25% of the structures (60% for the central region). The variability between possible structures that we find at the first order is hence very limited.

In contrast, the second-order structures (right panel) form a larger cloud, where all nodes are more spread out, revealing a larger variability in second-order HETNs for the nodes of all datasets (except for the primates dataset). Here, there is no particular region where the concentration of points is denser than in the others. This reveals that there are not many structures that are common to different nodes, as confirmed by observing the reported structures below: the most frequent HETN in each region covers less than the 25% of the total (less than the 5% for the outer region).

Finally, having analyzed the HETNS variability among datasets and between nodes, we can also study how individuals change their behavior over time. In Figure 6, instead of comparing different nodes, we compare each node with itself at different times. In particular, we divide the time series of interactions into daily intervals ($\left[T_{0},T_{1},\ldots ,T_{N}\right]$). The HETNs extracted for one node at $T_{0}$ are hence compared with those extracted for the same node in $T_{1}$; the same for $T_{1}$-$T_{2}$ up to $T_{N-1}$-$T_{N}$. The panels report the distribution of the distances computed for all the nodes considering all orders together, the purely first-order and the purely second-order, respectively. The difference between the overall behavior for first and second orders is even greater, with distances that pass from an average of around 0.24 for the first order only to an average of around 0.90 for the second order.

Taken together, our results show that a high variability between second-order structures (both among different nodes and among different temporal intervals of the same node) is always observed. This supports the idea that a higher-order analysis is hence needed to distinguish the social behaviors of different nodes and in different datasets. Focusing only on first-order structures is indeed limiting, because most of the extracted HETNs are trivial and common to all nodes and to all datasets, and do not allow revealing singular peculiarities. To clean the data from these trivial structures requires the use of a null model (as we did in the HETM analysis), which can however introduce biases. The second-order structures instead allow more refined analyses.

## 4. Conclusions

Considering the temporal dimension is fundamental to understand social interactions, and, in this sense, temporal networks provide a unique tool for their analysis. Nevertheless, capturing and modeling all the complexity of temporal interactions is an open question in network science.

To represent these temporal structures, egocentric temporal neighborhoods/motifs have emerged as a concise yet powerful tool for analyzing and modeling their dynamics. Focusing only on ego nodes and their connections allows us to analyze each node individually and represent the evolution of the network in a parsimonious way. In recent years, however, a new paradigm has emerged highlighting the role of higher-order structures in shaping the dynamics of social systems. In this approach, the emphasis is on correlations in the interactions between neighbors of the ego nodes: something that the original definition of temporal neighborhoods neglects in its construction [35]. Thus, the question of how relevant are higher-order interactions for the modeling of this class of dynamics naturally arose.

To answer this question, in this work, we used a generalization of temporal graphs to temporal hypergraphs, and we extended the concept of egocentric temporal neighborhoods to include second-order interactions—connections involving two neighbors of the ego node that are also interacting—for defining hyper egocentric temporal neighborhoods. This extension allowed the analysis of several datasets from face-to-face interactions covering different contexts and aggregation times. The datasets ranged from interactions between students in primary and high schools, to workplaces and hospitals, and baboons living in a research facility.

While first-order signatures were more abundant, as expected, second-order ones still played an important role. In fact, they were present in the most frequent structures in all datasets, with their signatures being statistically relevant even for larger aggregation times.

Interestingly, we also found that, at the whole hypergraph level, first-order structures were way more similar than second-order ones, with the latter showing a larger variability and diversity. This translates into most of the differences between hypergraphs being due to second-order HETMs.

This same result also holds if we zoom in at the individual node level: second-order structures are more sensitive to node dissimilarities. Moreover, inside each dataset, these distances between node signatures allowed better clustering of individuals according to their function; i.e., nodes with similar functions such as patients or administrative staff in the *Hospital* dataset showed similar signatures.

Finally, looking at distances between signatures of the same nodes at different periods in time, we demonstrated that second-order structures also showed a higher temporal variability.

Taken together, our results demonstrate that, while first-order HETMs provide a backbone for social interactions, second-order ones are fundamental for capturing their heterogeneities at all scales: the whole network, between nodes, and temporally, even at the individual level. This picture demonstrates, once more, the relevance of higher-order structures in shaping social and network dynamics in general.

However, the extension of egocentric temporal neighborhoods to second-order motifs comes at a cost. Second-order encodings, unlike first-order ones, are not unique, forcing us to test for isomorphic patterns; a major disadvantage that first-order analysis overcame. However, this limitation is mainly theoretical. For all the datasets and aggregation times considered, the computational costs of the isomorphism tests were almost negligible, with a running time for computing nodes’ signatures of a few minutes for the largest dataset and the longest aggregation time. A second limitation of our work is the fact that we limited our analyses to second-order motifs, instead of considering third- or even higher-order structures. However, their limited number suggests that most of the connections of the networks would have been covered by only considering first- and second-order HETMs.

In conclusion, with this work, we demonstrate that considering hyper egocentric temporal motifs in the analysis of temporal datasets is fundamental for capturing the intrinsic variability in social interactions.

We indeed observed non-trivial patterns in the way group interactions appear, in how they evolve in a short time range, and in how they are related to each other when taking place at the same time. The method that we propose, based on HETMs and HETNs, is able to simplify these pieces of information and encode them in binary signatures, which is useful for storing and comparing them, and which could be useful for further research. Future researchers could benefit from this method to help characterize their temporal higher-order data or to classify them into universality classes based on these signatures. Modeling and generation of synthetic surrogates is one direct and foreseeable application of this method that we envisage exploring in further work.

## Figures and Tables

**Figure 1 entropy-26-00256-f001:**
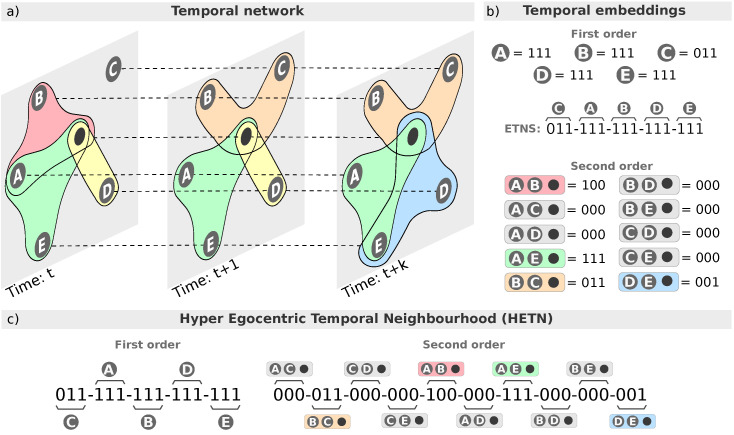
**The HETN-gen model:** (panel **a**) Schematization of a hyper egocentric temporal neighborhood with (k=2) and (panel **b**) its corresponding encoding at first and second order. (panel **c**) Hyper egocentric temporal neighborhood signature (HETNS) describing the HETN in (**a**).

**Figure 2 entropy-26-00256-f002:**
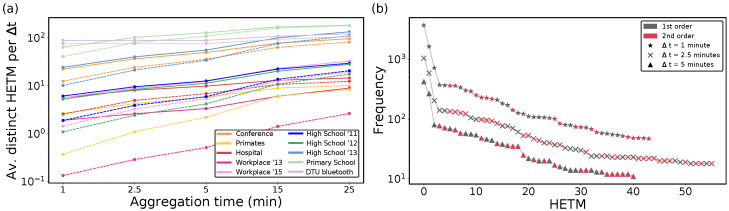
**HETM counts and abundance:** (panel **a**) Average number of distinct hyper egocentric temporal motifs (HETMs) per temporal layer in proximity contact data for different aggregations of time Δt and k=2. Continuous lines correspond to first- and second-order HETMs, and dashed lines to pure second-order HETMs. (panel **b**) Rank plot of the abundance of HETMs in the *Hospital* data (excluding the 20% less common HETMs) for different aggregation times. Points in grey indicate the first-order HETMs and points in red are second-order HETMs.

**Figure 3 entropy-26-00256-f003:**
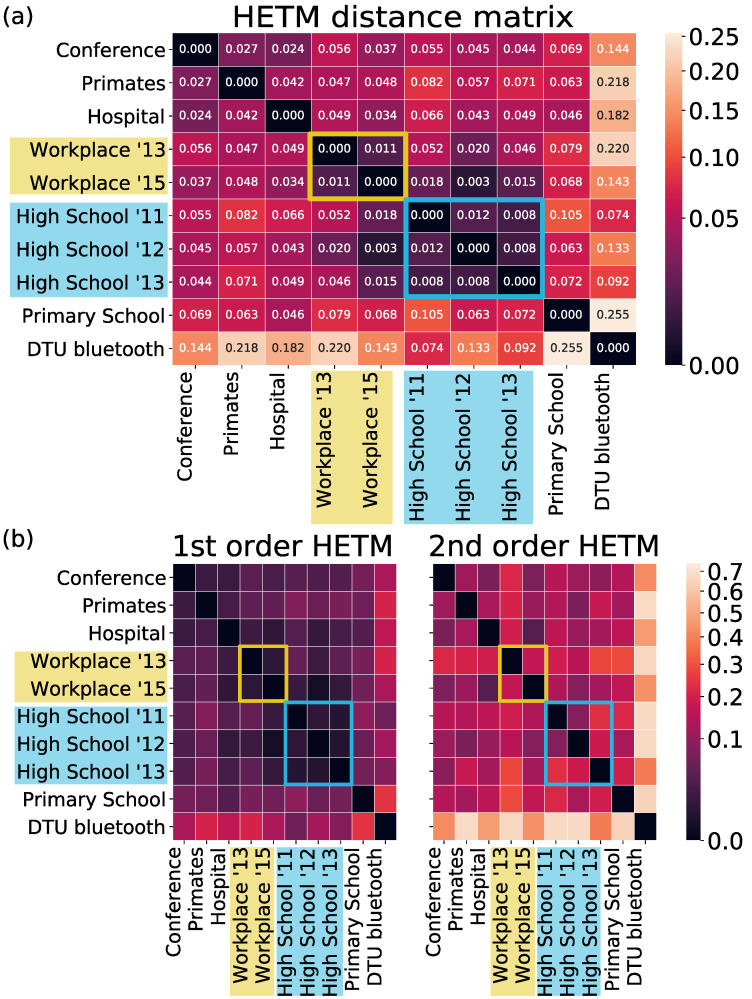
**HETM-based distances:** (panel **a**) Distance matrix among datasets based on HETMs. Results are shown for Δt=2.5 min and k=2. (panel **b**) Distances computed by only considering the first- or second-order HETMs, respectively. Color bars are represented in a logarithmic scale. In both panels, the highlighted measures correspond to equal settings sampled over different years.

**Figure 4 entropy-26-00256-f004:**
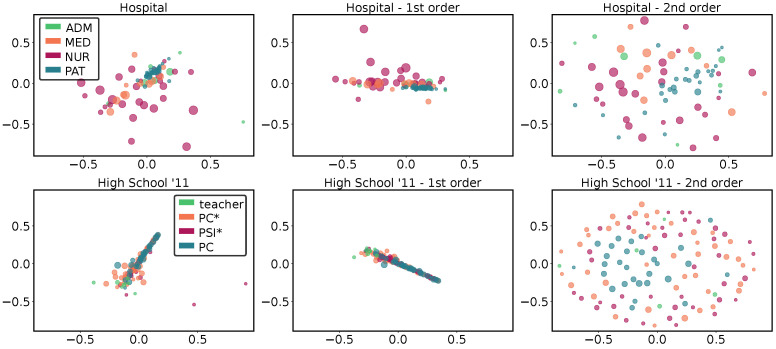
**HETN-based node distances for single networks:** Multidimensional scaling of HETN-based distances at single node level with Δt=2.5 min k=2 for two different proximity contact datasets: *Hospital* (**upper panels**) and *High School’11* (**lower panels**). We show the multidimensional scaling accounting for all the HETNs (**left panels**) and those obtained considering only first- (**central panels**) and second- (**right panels**) order HETNs. The size of the nodes is proportional to the number of different HETNs that each node presented. In the *Hospital* dataset (**upper panels**), the color of the nodes corresponds to the four different classes: administrative staff (green), medical doctor (orange), paramedical staff (purple), or patient (blue). In the *High School’11* dataset (**lower panels**), the color of the nodes depends on whether the node is a teacher (green) or a student belonging to three different classes: PC* (orange), PSI* (purple), or PC (blue).

**Figure 5 entropy-26-00256-f005:**
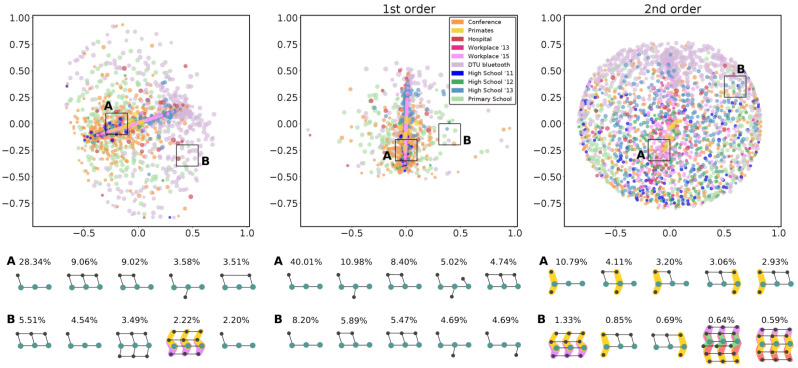
**HETN-based node distances in multiple networks:** (upper panels) Multidimensional scaling of HETN-based distances at single node level for Δt=2.5 min and k=2 for the 10 different types of social contact data considering both (**left**), first- (**middle**), and second- (**right**) order HETNs. (lower panels) Each row shows the five most frequent HETNs in their corresponding patch displayed in the cartography above. The percentage of each HETN indicates the percentage at which that signature appeared among all the HETNs of the nodes in the patch.

**Figure 6 entropy-26-00256-f006:**
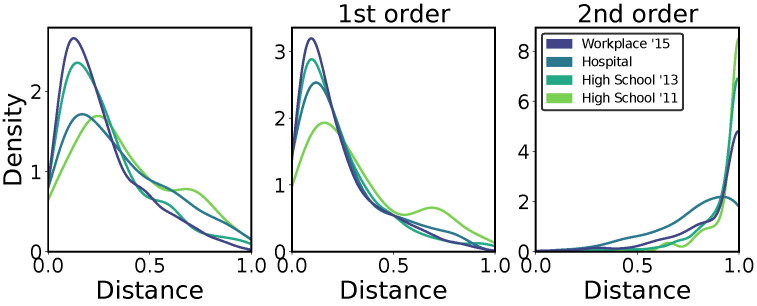
**Temporal distance distributions:** This is computed for every node with itself comparing their HETN embeddings for Δt=2.5 min and k=2. Distances are calculated evaluating the HETN embedding of each node in consecutive daily time windows for each dataset. Each *pdf* is normalized so that the total area of each histogram equals one.

**Table 1 entropy-26-00256-t001:** **Dataset statistics:** Number of nodes, number of edges, and duration (in days) for each network.

Name	|V|	|E|	Duration
Conference	403	70,261	1.32
Primates	13	63,095	28
High school 11	126	28,561	3.1
High school 12	180	45,047	8.44
High school 13	327	188,508	4.2
Primary school	242	125,773	1.35
Workplace 13	92	9827	11.4
Workplace 15	217	78,249	11.5
Hospital	75	32,424	4
DTU Bluetooth	656	717,669	7

## Data Availability

The data that support the findings of this study are openly available in http://www.sociopatterns.org/.

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
