# Peer review of "Patterns in Temporal Networks with Higher-Order Egocentric Structures"

_entropy, 2024, doi:10.3390/e26030256_

Round 1

Reviewer 1 Report

Comments and Suggestions for Authors

In their manuscript titled "Patterns in temporal networks with higher-order egocentric structures," the authors delve into the importance of integrating higher-order structures into the analysis of temporal networks, particularly for understanding social interactions. They introduce "hyper egocentric temporal motifs," which involve interactions between two neighbors of an ego node. Their findings reveal that while first-order interactions are prevalent, second-order interactions are also essential and demonstrate greater variability. Despite computational challenges, considering these structures enhances the modeling and comprehension of social dynamics.

However, the authors did not extensively investigate the impact of higher-order structures beyond second-order motifs. Additionally, they did not thoroughly explore the potential significance of third or higher-order interactions in shaping social dynamics, leaving questions unanswered about the complete spectrum of higher-order effects on temporal network dynamics.

Nevertheless, I find the work interesting and potentially suitable for publication after minor revisions.

It appears the authors did not provide physical interpretations for their obtained results. For example, in lines 256-257, where they mention a higher similarity between high schools and workplaces, no explanation is given for this observation. I suggest the authors include such interpretations throughout their study to help readers better understand their results.

Adding brief statements in the captions of tables and figures to highlight the key messages would be beneficial. Furthermore, the authors should review the manuscript for grammatical errors.

It would be advantageous to include a comparative figure with physical interpretations that compare first-order and second-order effects and their combined impact.

For an enhanced introduction, the authors may consider incorporating the following references:

- "Controlling species densities in structurally perturbed intransitive cycles with higher-order interactions." Chaos: An Interdisciplinary Journal of Nonlinear Science 32, no. 10 (2022).

- "Distance dependent competitive interactions in a frustrated network of mobile agents." IEEE Transactions on Network Science and Engineering 7, no. 4 (2020): 3159-3170.

- "Synchronization to extreme events in moving agents." New Journal of Physics 21, no. 7 (2019): 073048.

Comments on the Quality of English Language

Overall, the English language used in the manuscript is clear and understandable. The authors effectively convey their ideas and findings. However, there are a few areas where improvements can be made like some sentences could be rephrased for clarity and grammatical correctness.

Reviewer 2 Report

Comments and Suggestions for Authors

The authors examine hyper egocentric temporal neighbourhoods, which are networks viewed from the perspective of a node's neighbourhood (egocentric), but where interactions between nodes can be many-body (hyper-network/simplicial complex), and have a temporal aspect. The novelty stems from this temporal aspect, which is an interesting and important advance. They also study the model based on some real proximity contact networks (10 in total), which is an advantage.

The results are not controversial, since the authors simply attempt to define and study the model they introduce.

I think the results are scientifically sound, within reasonable limits.

I think the readers will be interested in how to best set up a situation like this, and how it works in practice.

If the paper could be improved, I'd say, go into more detail about what you observe. The conclusions are a bit vague. It may be better to not try and conclude new observations, simple to argue that you demonstrate how to use this method to reveal new structure, given the sophistication of the approach. Linking challenges with the datasets (things which we don't understand or can't explain) to this new modelling approach would be ideal but not necessary. Perhaps future researchers will find this interesting given the temporal aspects of their higher order data, and they can then progress the area in application to e.g. help classify or characterise their data effectively into universality classes, based on you signatures.

Perhaps explain the results cited in the literature about higher-order networks a bit more clearly. What are the big papers recently? Same with temporal networks. To huge fields, would be good to see the "state of the art" explained a bit, then show how "hyper egocentric temporal neighbourhoods" develop this academic investigation. A good and clear example of where all these things are critical (time, higher-order, and with interesting neighborhoods).

One missing citation, to be comprehensive, is a recent work of Giles and Bianconi on egocentric hypernetworks: Beyond the clustering coefficient: A topological analysis of node neighbourhoods in complex networks, Chaos, Solitons & Fractals: X, 1, 2019

which looks at ego-centric networks and their higher-order structure. Though, they do not look at the temporal aspect.

Overall, I say accept, given some minor revisions to the text to ensure exceptional clarity, readability, the extra citation, and any further minor modifications to the exposition to give it the best chance of seeing further research.

Author Response

See the attached document.

Round 2

Reviewer 2 Report

Comments and Suggestions for Authors

The paper has been revised effectively, and I recommend acceptance.